# A machine learning approach for rapid early detection of *Campylobacter* spp. using absorbance spectra collected from enrichment cultures

**Kefeng Zhang**[1]*, **Christelle Schang**[2], **Rebekah Henry**[2,3], **David McCarthy**[2,4]

**1** Water Research Centre (WRC), School of Civil and Environmental Engineering, UNSW Sydney, Sydney, New South Wales, Australia, **2** Department of Civil Engineering, Environmental and Public Health Microbiology Laboratory (EPHM Lab), Monash University, Melbourne, Victoria, Australia, **3** School of Public Health and Preventive Medicine, Monash University, Melbourne, Victoria, Australia, **4** School of Civil and Environmental Engineering, Queensland University of Technology, Brisbane, Queensland, Australia

* Kefeng.zhang@unsw.edu.au

**Data Availability Statement:** The data used in this study is shared in the following repository: https://doi.org/10.5281/zenodo.10547727

## Abstract

Enumeration of *Campylobacter* from environmental waters can be difficult due to its low concentrations, which can still pose a significant health risk. Spectrophotometry is an approach commonly used for fast detection of water-borne pollutants in water samples, but it has not been used for pathogen detection, which is commonly done through a laborious and time-consuming culture or qPCR Most Probable Number enumeration methods (i.e., MPN-PCR approaches). In this study, we proposed a new method, MPN-Spectro-ML, that can provide rapid evidence of *Campylobacter* detection and, hence, water concentrations. After an initial incubation, the samples were analysed using a spectrophotometer, and the spectrum data were used to train three machine learning (ML) models (*i.e.*, supported vector machine - SVM, logistic regression–LR, and random forest–RF). The trained models were used to predict the presence of *Campylobacter* in the enriched water samples and estimate the most probable number (MPN). Over 100 stormwater, river, and creek samples (including both fresh and brackish water) from rural and urban catchments were collected to test the accuracy of the MPN-Spectro-ML method under various scenarios and compared to a previously standardised MPN-PCR method. Differences in the spectrum were found between positive and negative control samples, with two distinctive absorbance peaks between 540-542nm and 575-576nm for positive samples. Further, the three ML models had similar performance irrespective of the scenario tested with average prediction accuracy (ACC) and false negative rates at 0.763 and 13.8%, respectively. However, the predicted MPN of *Campylobacter* from the new method varied from the traditional MPN-PCR method, with a maximum Nash-Sutcliffe coefficient of 0.44 for the urban catchment dataset. Nevertheless, the MPN values based on these two methods were still comparable, considering the confidence intervals and large uncertainties associated with MPN estimation. The study reveals the potential of this novel approach for providing interim evidence of the presence and levels of *Campylobacter* within

**Funding:** The first and corresponding author (Kefeng Zhang) is supported by Australian Research Council Discovery Early Career Researcher Award (ARC DECRA, DE210101155). The data collected and used in this paper were collected as part of two different Australian Research Council Linkage Projects (LP120100718, LP160100408) and another ARC DECRA (DE140100524). The authors would like to acknowledge Melbourne Water and EPA Victoria for co-funding of the ARC LPs. The funders had no role in study design, data collection and analysis, decision to publish, or preparation of the manuscript.

**Competing interests:** The authors have declared that no competing interests exist.

environmental water bodies. This, in turn, decreases the time from risk detection to management for the benefit of public health.

## 1. Introduction

*Campylobacteriosis* is a zoonosis introduced that is transmitted through contact with faecal material primarily derived from bovine and avian sources. Current WHO figures suggest that *Campylobacter* is the leading cause of diarrheal disease in industrialized nations with annually more than 60,000 and 17,000 confirmed cases reported respectively in the United Kingdom (UK) and Australia alone [1, 2]. From these, it is estimated that between 10%-30% are due to environmental exposure pathways [3, 4]. What makes *Campylobacter* so dangerous is that it can cause explosive, unpredicted outbreaks with the potential to affect everyone within the catchment [5, 6]. For example, in 2016, 5500 people (40% of the community) were infected with *Campylobacter* after consuming contaminated drinking water in Havelock North (New Zealand) [7]. Thus, testing the presence of *Campylobacter* is necessary for not only understanding transmission pathways, but its subsequent mitigation in the environment to the benefit of public health.

Enumeration of *Campylobacter* from complex source samples can be difficult. Isolation from water samples is particularly problematic, as they are usually present at low concentrations within these microbially complex environments [8]. Culture-based methods for the enumeration and isolation of *Campylobacter* from waters have been optimised (Standardization ISO, 2005). However, these procedures can be time-consuming and expensive, requiring filtration, selective enrichment, isolation, and biochemical confirmation (totaling up to ~9 days to report). A modified Most-Probable Number (MPN)-PCR method is described in Henry, Schang [9], evaluated by analysing 147 estuarine samples collected over a 2-year period, demonstrated that the intra-laboratory performance of an MPN-PCR approach was superior to that of the Australian/New Zealand Standard (AS/NZS) ($\sigma = 0.7912$, $P < 0.001$; $\kappa = 0.701$, $P < 0.001$) with an overall diagnostic accuracy of ~94% [10]This method reduced the reporting time to 4 days instead of the standard 9 days. However, both the traditi

onal culture-based method and the modified MPN-PCR method remain expensive, requiring specialised equipment and expertise. Therefore, cheaper and technically more accessible methods are still required.

With the rapid development of sensor technologies, optical techniques are now commonly used for the fast detection of waterborne pollutants. These include UV–Vis (ultraviolet–visible) spectrophotometry, or near-infrared spectroscopy NIR to characterise pollution levels in drinking water and wastewater systems [11, 12]. Further, optical density, or absorbance, has been widely applied for the estimation of bacterial concentrations in growth media and is often used in water analysis standards around the world [13–15]. However, these protocols frequently use a single wavelength to investigate mono-cultures within specific growth media. Rapid techniques such as biosensors have also been developed for a range of organisms [16, 17]. However, to our knowledge, no studies have investigated comparable methods to detect and predict the concentration of a waterborne pathogen in a complex matrix, such as those represented within environmental waters (*e.g.*, streams, rivers, estuaries).

Machine learning approaches have been used as efficient tools to establish the relationships between spectral results and the continuous monitoring of water quality. For example, Carreres-Prieto, García [12] developed different regression models, such as multivariate linear

regressions and machine learning genetic algorithms to estimate sewage water quality from UV-Vis spectrum data. Arnon, Ezra [18] proposed a new scheme for early detection of contaminant events in the water supply system through real-time UV-spectrophotometry, which applied a machine learning method to set contamination alarms. They found that the models required significant training with a defined dataset containing high variability (that can represent all water sources) to achieve significant detection rates while maintaining low levels of false positives. These methods, however, are commonly applied in wastewater or drinking water systems and focused on bulk parameters (*e.g.*, biological oxidation demand, chemical oxidation demand, total suspended solids, total phosphorous and nitrogen species [12], and organic contaminants [18]). However, there is a dearth of relevant applications in environmental waters despite increasing concerns about health risks associated with exposure to pathogens during recreational use (*e.g.*, swimming and boating). Consequently, the potential of using spectrophotometry coupled with machine learning models to predict the presence of pathogens (e.g., *Campylobacter*) in these complex matrices remains unexplored.

This study proposed the application of a new method, named MPN-Spectro-ML, that can provide a fast turnaround time when detecting and enumerating *Campylobacter*, as an alternative, or a precursor, to the traditional MPN-PCR method. The procedure applies a spectrophotometer to analyse the initially incubated sample with machine learning models to process the spectrum data to predict *Campylobacter* presence within enriched water samples. These values are then utilised to estimate the Most Probable Number (MPN) within the water samples. The described study applied water from a range of urban and rural catchments in Melbourne, Australia, with the specific objectives of:

- investigate the absorbance spectrum of enrichment cultures that are positive or negative for *Campylobacter*, where those enrichment cultures are derived from a variety of water sources, *i.e.*, stormwater, river, and creek samples (including both fresh and brackish water),

- test and compare three machine learning approaches (logistic regression - LR, random forest - RF, supported vector machine - SVM) in predicting the presence of *Campylobacter* by using the spectrum data under various scenarios and

- evaluate the new MPN-Spectro-ML method's capability in predicting the presence/absence of *Campylobacte*r and estimating the concentration of *Campylobacter* (MPN/L) within the samples, compared to the traditional MPN-PCR method.

These results of this work demonstrate the potential of spectrophotometry for interim reporting of the presence and concentration of *Campylobacter* in water systems. This could potentially pave the way to reduce turnaround times and associated healthcare costs as a result of delayed risk reporting. This will enable more timely and effective reporting of public health risks associated with aquatic recreation at monitoring sites.

## 2. Methodology

Fig 1 presents the overall methodology of this study. Section 2.1 details the sample collection process. Section 2.2 introduces the traditional *Campylobacter* analysis approach, *i.e.*, *MPN-PCR*, which involved sample preparation, inoculation, PCR analysis and MPN estimation. Section 2.3 presents the new *MPN-Spectro-ML* method which is based on spectrophotometry analysis and machine learning model.

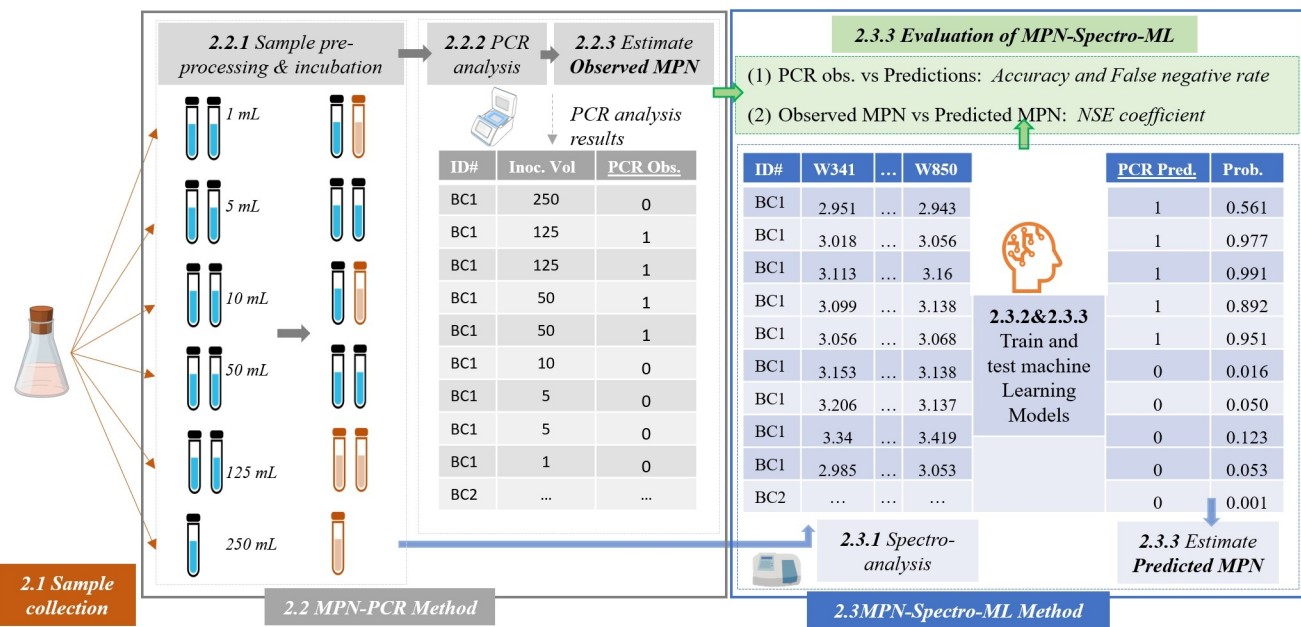

**Fig 1. Overview of the methodology from sample collection, culturing, molecular and spectrophotometric quantification.**

## 2.1. Sampling collection

Water samples were collected at three creeks with various catchment characteristics in Melbourne, Australia (Table 1). Water was collected in 2 L polyethylene terephthalate containers rinsed with a minimum of 1 L of source water prior to collection, as previously described in Henry, Schang [9]. Samples were collected 3 m perpendicular to the nearest bank at an approximate depth of 0.15 m. Sampling days were selected to incorporate variable climatic and hydrological conditions with rain event samples collected using a flow-weighted strategy (McCarthy et al., 2008). The permits for accessing all the sites were acquired from the asset owner Melbourne Water.

**Table 1. Summary of the number of samples collected from the three different sites, the number of tubes analysed by PCR, and the number of positive and negative observations out of the total number of samples tested using the PCR-MPN approach.**

| Type of dataset | | Sites characteristics | No. of samples | No. of PCR analysis (tube) | Pos: negative obs. (based on PCR analysis) |
|---|---|---|---|---|---|
| Controls ($C_{all}$) | | n/a | n/a | 177 | 44 pos[1], 133 negatives[2] |
| Water Samples ($W_{all}$) | Rural catchment ($W_{Rural}$) | 4 sites, fresh water | 24 | 264 | 38 pos, 226 neg |
| | Urban catchment ($W_{Urban}$) | 2 sites, fresh water | 16 | 187 | 96 pos, 91 neg |
| | Mixed rural and urban catchment ($W_{Mix}$) | 8 sites, fresh and brackish water | 99 | 1089 | 390 pos, 698 neg |
| **Total** | | | 139 | 1717 | 568 pos, 1148 neg |

[1] Positive control sample = *Campylobacter jejuni* control.

[2] Negative control samples include *E. coli* controls, no antibiotic control (NAB) and sterile water controls.

## 2.2. MPN-PCR method

Water samples were analysed for *Campylobacter spp.* using the MPN-PCR method described in Henry, Schang [9], following two steps: sample pre-processing and initial incubation (Section 2.2.1) and PCR analysis using the enriched culture (Section 2.2.2). PCR results were applied to estimate the most probable number (MPN) (Section 2.2.3)

**2.2.1. Sample pre-processing and initial incubation.**   Water sample aliquots were filtered through a 0.45 µm cellulose membrane before being introduced into 25mL of Preston broth (Nutrient Broth No. 2, Oxoid, United Kingdom) containing 0.05% Horse Blood (AEB)). Volumes ≤1 mL were directly introduced into 10 mL of Preston broth. A total of 11 tubes per sample were processed with three main filtrate regimes applied (as illustrated in **Fig 1**). These were:

(A) $1 \times 250$ mL, $1 \times 125$ mL, $1 \times 50$ mL, $1 \times 10$ mL, $2 \times 5$ mL and $5 \times 1$ mL, (B) $1 \times 250$ mL, $2 \times 125$ mL, $2 \times 50$ mL, $2 \times 10$ mL, $2 \times 5$ mL and $2 \times 1$ mL, (C) $1 \times 250$ mL, $1 \times 125$ mL, $1 \times 50$ mL, $1 \times 10$ mL, $2 \times 5$ mL and $5 \times 1$ mL, (D) $1 \times 250$ mL, $2 \times 125$ mL, $2 \times 50$ mL, $2 \times 10$ mL, $2 \times 5$ mL, $3 \times 1$ mL, $2 \times 0.1$ mL. Post-filtration onto 0.45 µm cellulose nitrate filters, tubes were resuscitated for 2hrs at 37˚C before 100 µL (or 50 µL for the 10 mL tubes) of *Campylobacter* selective supplement (Oxoid, United Kingdom) were added into each inoculum. Samples were then incubated for 48 hrs at 42˚C in microaerophilic conditions (85% $N_2$, 10% $CO_2$, and 5% $O_2$).

**2.2.2. PCR analysis.**   After 48 hrs incubation, a total of 2 µL of the enriched culture was diluted into 20 µL of UltraPure DNase RNase free distilled water (Invitrogen, USA) and stored at -20˚C for a minimum of 16hrs. The samples were then tested by qPCR using the method described in Henry et al. (2015). No antibiotic negative enrichment controls were included to ensure no media contamination. *Campylobacter jejuni*, *E. coli*, no antibiotic and DNA-free water contamination controls were conducted with each assay as outlined in AS/NZS [10]. Details for the primers, mastermix and qPCR cycling conditions are described in Henry et al. (2005). Briefly, the qPCR analysis used Biorad SsoFast Evagreen (BIORAD) mastermix as per manufacturer's specified cycling conditions. *Campylobacter* spp. primers were obtained from IDT, with qPCR conducted using a CFX96 thermocycler (BIORAD). Positive and negative control samples were conducted in duplicate as described in Henry et al. 2015.

**2.2.3. Most probable number (MPN) estimation.**   The PCR analysis results from all 11 tubes for each sample (*i.e.* positive or negative *Campylobacter* presence in each tube) were then used to estimate the Most Probable Number (MPN) based on Briones and Reichardt [19] and Garthright and Blodgett [20]. The MPN method permits the estimation of population density without an actual count of single cells or colonies. MPN provides a quantitative estimate of bacterial concentration, which is more informative for assessing contamination levels and potential health risks. MPN estimation is based on a determination of the presence or absence of microorganisms in several individual proportions of each of several dilutions of a sample (as introduced in **Fig 1**, Section 2.2.1 and 2.2.2). Based on the number of positive and negative tubes receiving a known quantity of inoculum, the MPN of microorganisms can be estimated by applying probability theory [21]. This theory calculates the probability that a particular tube among replicates will contain at least one bacterium (in this case, *Campylobacter*), indicated by a positive response after incubation. We can determine the probability of each pattern by considering all possible combinations over a range of bacterial numbers (n). From the resulting bar graph of these probabilities versus n, we can identify the Most Probable Number (MPN) - the value of n for the highest bar divided by the total volume in the test setup - and its occurrence probability [22]. The 95% confidence interval was also estimated using Haldane's approximation [23].

## 2.3. MPN-Spectro-ML method

After 48 hrs of incubation (Section 2.2.1), a sub-sample was also collected for spectrophotometry analysis (Section 2.3.1). The collected spectral data was used to train three independent machine learning models to predict the presence of *Campylobacter* spp. in each tube (Section 2.3.2). The prediction results were then used to estimate the predicted MPN (introduced in Section 2.3.3).

**2.3.1. Spectrophotometry analysis.** After incubation, 100 μL of each Preston broth tube for each of the samples was transferred into a tissue culture 96 wells microplate (Falcon) and analyzed by a Multiskan Sky spectrophotometer (Thermo Fisher Scientific). The absorbance spectrum of each well was scanned for wavelengths between 220nm and 850nm, corresponding to the UV-vis spectrum. Using the SkanIt software (Thermofisher), the absorbance spectrum was corrected by applying the blank subtraction function. The plate used to measure the absorbance did not pass the UV spectrum, and therefore wavelengths 220nm to 340nm were removed from the analysis.

Pearson correlation analysis was performed by using IBM SPSS Statistics software to understand the linear relationships between the absorbance data (full spectrum from 453 nm to 850 nm) and the presence/absence of *Campylobacter* based on PCR analysis results. Visual comparisons of the absorbance spectrum were then made for the positive control and negative control samples, as well as the water samples (which were further separated into positive and negative samples based on PCR results). The analysis of water samples was also conducted at the overall level (all sites combined) and the site level. This was done to gain a visual indication as to whether specific wavelengths could be linked to the presence of *Campylobacter* in the tubes.

**2.3.2. Machine learning models and preliminary testings.** *2.3.2.1. Machine learning (ML) models*. Three common ML classification approaches were applied in this study to predict the presence of *Campylobacter* in the incubated tubes (*i.e.* positive/negative or probability) by using the absorbance spectrum data. The first ML method used was logistic regression (or logit regression, **LR**), a statistical model that has been used for water quality simulations (e.g., [24]). In this study, we used LR to find the probability of *Campylobacteria* presence (*p*) in the collected water samples. It learns a linear relationship between independent variables (*i.e.*, in this case, absorbance at different wavelengths) and the log-odds (the ratios of the probabilities of the event happening to it not happening, i.e., *log(p/(1-p))*) from the given dataset [24, 25, 26]. The second approach was Support Vector Machines (**SVM**), which is a common machine learning technique for classification [27, 28] and has been commonly applied to predict water quality in freshwater bodies [29, 30]. Briefly, SVM employs a N-dimensional hyperplane to separate the datasets into two categories using suitable kernel functions, such as linear, Gaussian, polynomial, etc. It follows the principle of Structural Risk Minimization (SRM), minimising the expected error of a learning tool and thus reduces the problem of overfitting, making it capable of dealing with a large number of input dimensions (*e.g.*, in this study, wavelength data) with a relatively low level of computational complexity. The third ML approach used was random forest (**RF**), which is an ensemble method that trains many decision trees in parallel with bootstrapping followed by aggregation [31]. In RF, each individual tree is constructed by a random subset of training dataset based on different subsets of available variables (in this case, the wavelengths). Each node in RF is split using the best among a subset of wavelengths randomly chosen at the node, which is different from the decision tree method which uses all the data and the best variable for splitting the data. By aggregating many decision trees in the forest, RF can limit the overfitting, variance, and error caused due to bias.

*2.3.2.2. Preliminary testing*. The three ML models were applied using the relevant tools within the open-source library *Scikit-learn* (Python 3), *i.e.*, *sklearn.linear_model*.

*LogisticRegression*, *sklearn.svm.SVC*, and *sklearn.ensemble.RandomForestClassifier*. To test these models, a set of hyperparameters needed to be determined; therefore, preliminary modeling exercises were conducted to determine these parameters. Briefly, 1,000 runs (training and testing of each model with 80–20 random split of all the data for training and testing) were conducted firstly to gauge the range and sensitivity of the hyperparameters that were thought to impact the model performance, followed by another 1,000 runs to determine the impact of hyperparameters. Most of the hyperparameters were insensitive, and thus, the default values were set. Supporting Information S1 File S1 Table summarises the ranges of these hyperparameters tested and the final selected values for each hyperparameter.

**2.3.3. Evaluation of the MPN-Spectro-ML method.** *2.3.3.1. Testing scenarios for PCR predictions.* All the datasets shown in Table 1 were grouped into sub-datasets: all control sample dataset ($C_{all}$), from which a subset of these data with an even number of positive and negative controls was created ($C_{even}$); the dataset with all water samples ($W_{All}$) was further separated into: a rural catchment subset ($W_{Rural}$), an urban catchment subset ($W_{Urban}$) and a mixed catchment subset ($W_{Mix}$). Based on these sub-datasets, a total of nine different testing scenarios were designed (**Table 2**):

- **Scenarios 1–2**: Use $C_{all}$ (or $C_{even}$) for model training and $W_{all}$ for model testing. This was to investigate whether pure control samples can be prepared and measured in the laboratory to train the model and use it directly for the prediction of real water samples (Scenario 1: ***Train_$C_{all}$ + Test_$W_{all}$***), and to test whether an uneven number of control samples can have an impact on the testing results (Scenario 2: ***Train_$C_{Even}$ + Test_ $W_{all}$***),

**Table 2. Design of scenarios for testing the MPN-Spectro-ML method.**

| Scenario | | Train and test dataset | Objective |
|---|---|---|---|
| 1 | $C_{all} + W_{all}$ | Train: all control samples | Test if lab control samples can be used to train the model and use directly for the real water samples |
| | | Test: all water samples | |
| 2 | $C_{even} + W_{all}$ | Train: even number of positive and negative control samples | Test if uneven control sample numbers can have an impact on the testing results as compared to *Scenario 1* |
| | | Test: all water sample | |
| 3 | $W_{all,80} + W_{all,20}$ | Train: 80% all water samples | Test if the model can be trained and tested just using all different environmental water samples (*i.e.*, without the need for lab control samples) |
| | | Test: 20% all water samples | |
| 4 | $C_{all} + W_{Rural}$ | Train: all control samples | Test specific catchments, using similar approaches as *Scenario 1* (using |
| | | Test: all rural catchment samples | |
| 5 | $W_{Rural,80} + W_{Rural,20}$ | Train: 80% Rural catchment samples | controls for training), and *Scenario 3* (only using water samples for training |
| | | Test: 20% Rural catchment samples | |
| 6 | $C_{all} + W_{Urban}$ | Train: all control samples | and testing) |
| | | Test: all urban catchment samples | |
| 7 | $W_{Urban,80} + W_{Urban,20}$ | Train: 80% urban samples | |
| | | Test: 20% urban samples | |
| 8 | $C_{all} + W_{Mix}$ | Train: all control samples | |
| | | Test: all mix catchment samples | |
| 9 | $W_{Mix,80} + W_{Mix,20}$ | Train: 80% mix catchment samples | |
| | | Test: 20% mix catchment samples | |

- **Scenario 3**: Used the $W_{all}$ sub-dataset only for model training and testing, with an 80–20 split (i.e., randomly select 80% of the dataset for model training and use the rest for testing) (Scenario 3: $\boldsymbol{Train\_W_{80} + Test\_W_{20}}$ Scenario). This was used as a comparison to the previous scenario, and

- **Scenarios 4–9**: Focused on catchment-specific datasets–to perform the previous two scenarios for each catchment dataset separately. For example, using the rural catchment samples: trained with $C_{all}$ and used $W_{Rural}$ for testing (Scenario 4: $\boldsymbol{Train\_C_{all} + Test\_W_{Rural}}$); trained and tested using $W_{Rural}$ with 80–20 split (Scenario 5: $\boldsymbol{Train\_W_{Rural\_80} + Test\_W_{Rural\_20}}$ Scenario). This was to understand whether there is a need to train the method to particular catchment contexts.

In all these scenarios, the three ML models were run five repeated times to account for model variation. The model performance was evaluated by Confusion Matrix, based on which the *Accuracy* (*ACC* = (true positive + true negative) / total population) and *False Negative Rate* (*FNR*, or called *miss rate* = false negative / the number of real positive cases in the population) were calculated to evaluate and compare the performance of these models.

*2.3.3.2. Testing for MPN predictions.* The predicted binary results (*i.e.*, positive or negative *Campylobacter* presence) from the spectrum data of all 11 tubes for each sample and ML models were used to estimate the MPN according to the same methods in 2.2.3. Three ML approaches provide probability estimations, from which binary output is generated using a typical threshold of 0.5. Therefore, in addition to the MPN estimations based on binary predictions, we also considered the probability estimates to assess their potential for improving MPN estimation accuracy, i.e., the binary values were replaced with the probability estimates when computing MPN.

Nash-Sutcliffe efficiency (NSE) coefficient (Nash and Sutcliffe, 1970), which is widely used for the assessment of water quality models (e.g., [32–34]), was used in this study to evaluate the ability of MPN-Spectro-ML method in predicting MPN. The NSE is calculated using the Eq (1).

$$NSE = 1 - \frac{\sum_{i=1}^{n} (o_i - P_i)^2}{\sum_{i=1}^{n} (o_i - \bar{o})^2} \tag{1}$$

$O_i$ is the MPN values estimated from the MPN-PCR method (considered as the observed value); $P_i$ is the MPN values estimated from the MPN-Spectro-ML method (considered as the predicted value). $\bar{O}$ is the mean of the observed values (i.e., all MPN values from the MPN-PCR method). The NSE ranges from $-\infty$ to 1, with 1 indicating a perfect match between observed and predicted values. When NSE equals zero, the predictive power is equivalent to simply using the average of the observed values as the prediction for all time steps, while negative NSE values indicate that the model predictions are worse than using the mean of the observed data. Zhang, Randelovic [35] suggested that NSE values greater than 0.3 indicate moderate model performance, while NSE values less than 0.3 indicate poor model performance.

## 3. Results and discussion

Visual observation of enrichment cultures after 48 hrs of incubation exhibited distinctive changes in revealed media colouration, which appeared to be specific to certain samples and sub-samples. It was therefore hypothesised that observed differences may be directly linked to the growth of *Campylobacter*, rather than other enriched microorganisms. This intriguing finding raised the possibility of using spectrophotometric methods to predict the presence or absence of *Campylobacter* following the initial incubation period.

## 3.1. Characteristics of spectrum data

Initial Pearson correlation analysis of absorbance spectrum data with PCR analysis results indicated that the absorbance of 195 wavelengths had significant correlations to the presence of *Campylobacter* ($p < 0.01$) (refer to Supplementary Information–correlation results). The absorbance of 56 wavelengths (all in the range of 531 nm to 586 nm) having $R$ values over 0.40, indicated a relatively strong correlation. The highest correlated wavelength included wavelengths between 573 and 578 nm, with R values of 0.53. Indeed, previous studies also identified that for other pollutants, a range of different wavelengths relevant to their pollution levels, *e.g.*, based on statistical models (*i.e.*, genetic algorithms), Carreres-Prieto, García [12] found that eight different wavelengths were most relevant to COD (chemical oxidation demand) concentrations, while five other different wavelengths were most relevant to TSS (total suspended solids) concentrations in wastewater samples. This finding further supports the use of multiple wavelengths across the whole spectrum to increase the accuracy of prediction associated with the presence or absence of *Campylobacter* in Preston broth.

Visual differences in the spectral absorbance between 400nm and 850nm are illustrated in Fig 2. Specifically, the results revealed the presence of two distinctive local peaks in the positive control samples at 540-542nm and 575-576nm, which agreed with the correlation analysis results. In contrast, most negative control samples had no observable absorbance peaks within this range (*i.e.*, 92% of the samples; refer to Supporting Information S1 File S2 Table for details). However, two small local peaks at wavelengths of ~500 nm and 635 nm, respectively, could be identified within these samples (**Fig 2A and 2B**). It is promising that under ideal conditions (*i.e.*, controls prepared in the lab free from other microbes or pollutants), the presence of *Campylobacter* in Preston broth after incubation showed distinct characteristics of spectrum absorbance between positive and negative control samples. For environmental water samples, most of the positive samples using the MPN-PCR method displayed the two distinctive two peaks observed within the positive control (average 95% of the samples across all sites; **Fig 2C**). In contrast, negative water samples using the MPN-PCR method also displayed peaks comparable to positive samples at 540-542nm and 575-576nm (average 42.4% of all negative samples, **Fig 2D**). This was observed particularly within samples collected from the Rural catchment (63.3% of the samples; **Fig 2F**), a high-quality drinking water catchment. These changes may be directly associated with differences in nutrient usage by microbiota within the enriched samples, which have been previously observed to be highly variable and not specific to *Campylobacter* spp. [36]. Therefore, it may be of interest to investigate this phenomenon further within similar freshwater contexts to define the microbial source of this interference. These results provided the impetus to explore the use of machine learning approaches to further analyse the spectrum data.

## 3.2. Performance of the MPN-Spectro-ML method in predicting *Campylobacter* presence

The performance of the three ML models (SVM, logistic, and RF) had no significant differences ($p < 0.05$) in predicting the presence of *Campylobacter*, with an overall accuracy (ACC) of $0.728 \pm 0.118$ (**Table 3**). However, the false negative rates (FNR) of SVM and logistic models (average 9.7%) were comparably lower than RF (average 23.3%). By comparing the prediction results (presence of *Campylobacter*) of individual enrichment cultures, it was observed that there was > 90% similarity for the binary predictions from SVM and logistic models, as shown in the confusion matrix (**Fig 3**, Scenario 1 as an example). These results highlighted that all the models could learn from the given data to provide early predictions, but with RF giving higher FNR. These FNR results were also comparable to similar studies on early contamination

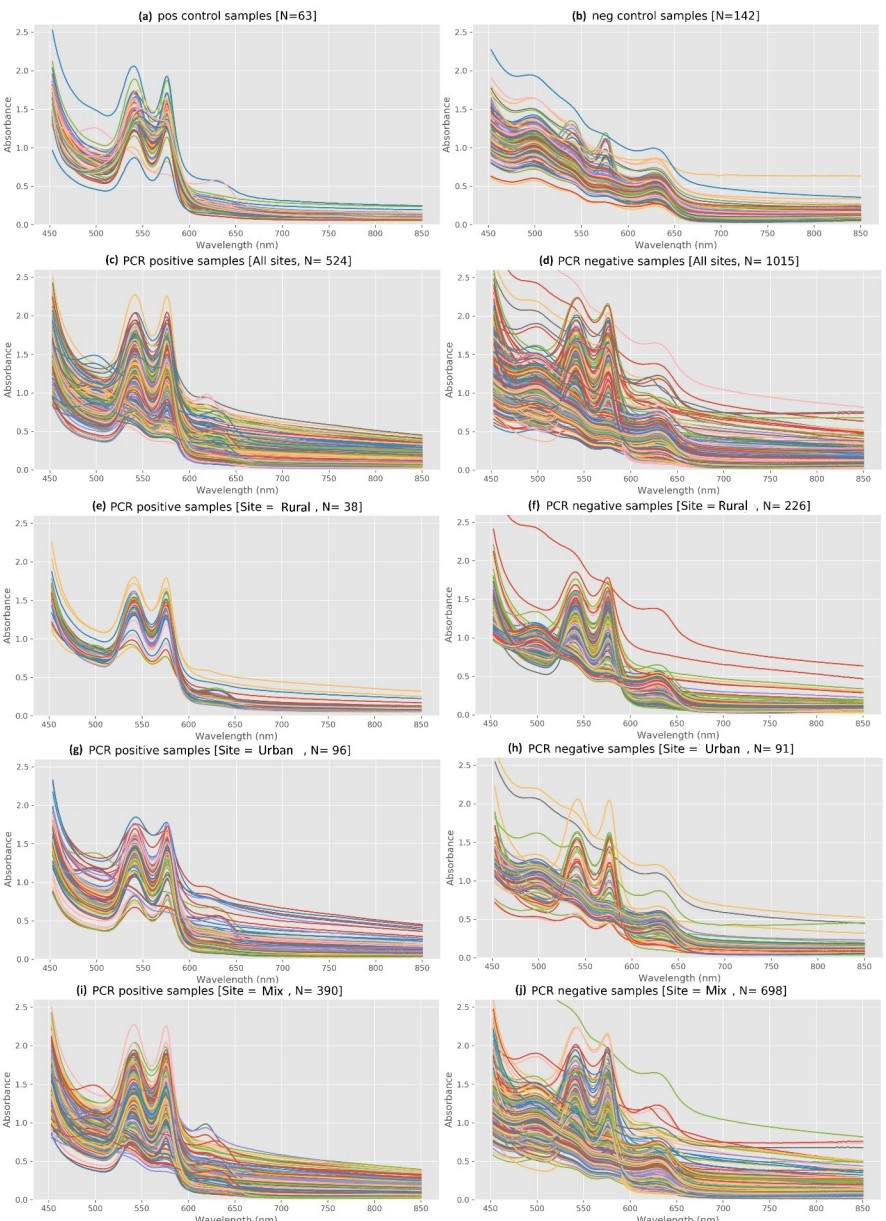

**Fig 2.** Absorbance spectrum of positive and negative control (a-b), and PCR positive and negative samples from all sites combined and the three study sites between wavelength 450 and 800 nm (c-j). The wavelength < 450 nm was not shown as they have variations with no observable difference between all samples.

detection in drinking water using UV-Spectrophotometry, *e.g.*, based on various datasets, Arnon, Ezra [18] used SVM to predict contamination events and found that the false negative rates (actual contaminated water predicted to be potable water) varied from only 1.42% to almost 28.8%. Thus, though it was noted that accuracies could be improved (*i.e.*, to over 0.90), these were considered sufficient in provisioning an early indicator of the potential risk of *Campylobacter* in water, which would then require secondary confirmations.

By comparing test **Scenario 1 $C_{all} + W_{all}$** (44 positive and 133 negative controls) and **Scenario 2 $C_{even} + W_{all}$** (44 positive and negative controls), no significant difference was observed

**Table 3. Summary of the model performance (based on test) under different test scenarios.** The accuracy and false negative values from the training phase are presented in the Supporting Information S1 File S3 Table.

| Test scenario | | Accuracy | | | False negative rate (FNR) | | |
|---|---|---|---|---|---|---|---|
| | | SVM | Logistic | RF | SVM | Logistic | RF |
| 1 | $C_{all} + W_{all}$ | 0.726 | 0.720 | 0.744 | 10.5% | 10.3% | 15.5% |
| 2 | $C_{even} + W_{all}$ | 0.728 | 0.730 | 0.737 | 10.7% | 10.5% | 13.2% |
| 3 | $W_{all,80} + W_{all,20}$ | 0.712 | 0.731 | 0.749 | 12.6% | 13.6% | 24.3% |
| | | (0.051)* | (0.043) | (0.053) | (3.1%) | (3.1%) | (6.4%) |
| 4 | $C_{all} + W_{Rural}$ | 0.466 | 0.451 | 0.519 | 7.9% | 7.9% | 13.2% |
| 5 | $W_{Rural,80} + W_{Rural,20}$ | 0.482 | 0.636 | 0.814 | 0.0% | 10.8% | 78.3% |
| | | (0.159) | (0.072) | (0.049) | (0.0%) | (24.1%) | (22.5%) |
| 6 | $C_{all} + W_{Urban}$ | 0.861 | 0.856 | 0.840 | 5.2% | 6.3% | 12.5% |
| 7 | $W_{Urban,80} + W_{Urban,20}$ | 0.836 | 0.836 | 0.752 | 7.4% | 11.0% | 16.4% |
| | | (0.046) | (0.041) | (0.050) | (2.0%) | (3.8%) | (6.1%) |
| 8 | $C_{all} + W_{Mix}$ | 0.766 | 0.762 | 0.780 | 12.1% | 11.5% | 16.9% |
| 9 | $W_{Mix,80} + W_{Mix,20}$ | 0.794 | 0.808 | 0.818 | 12.6% | 13.0% | 19.2% |
| | | (0.007) | (0.008) | (0.021) | (2.8%) | (3.8%) | (5.0%) |

Note * the numbers/percentages in the brackets represent the standard deviation from five repeated runs. This was only presented for the scenarios (3, 5, 7, and 9) that had a random split of 80–20 in the dataset for training and testing, while for the other scenarios the five repeated runs led to almost identical results.

($p<0.01$, **Table 3**). This was supported by the findings presented in **Fig 4**, where the percentages of true positive (TP), true negative (TN), false positive (FP), and false negative (FN) were found to be identical between these two scenarios. This indicated that having an uneven number of positive and negative controls for training had a negligible impact on the overall model testing results. Further, if only water samples were used for both training and testing (**Scenario 3 $W_{all,80} + W_{all,20}$**), although the same level of accuracies could be achieved (0.712–0.749), the levels of FNR were observed to be higher (average = 16.8%) in comparison to simulations where control data where integrated as part of the training dataset in Scenario 1 and 2 (11.8%). This highlighted that laboratory-prepared control samples were sufficient to train the ML models before applying the trained models for predicting *Campylobacter* in environmental water samples.

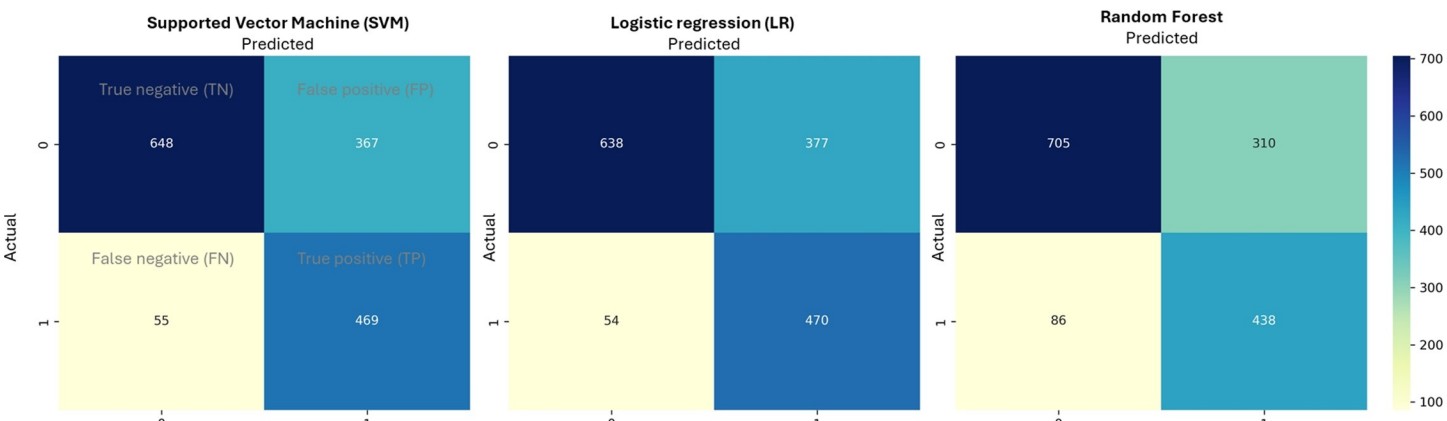

**Fig 3. Confusion matrix of the models testing results from Scenario 1 ($C_{all} + W_{all}$, using all the control samples for training and all water samples for testing).** ACC = (TN+TP)/(TN+TP+FN+FP), and FNR = FN/(FN+TP).

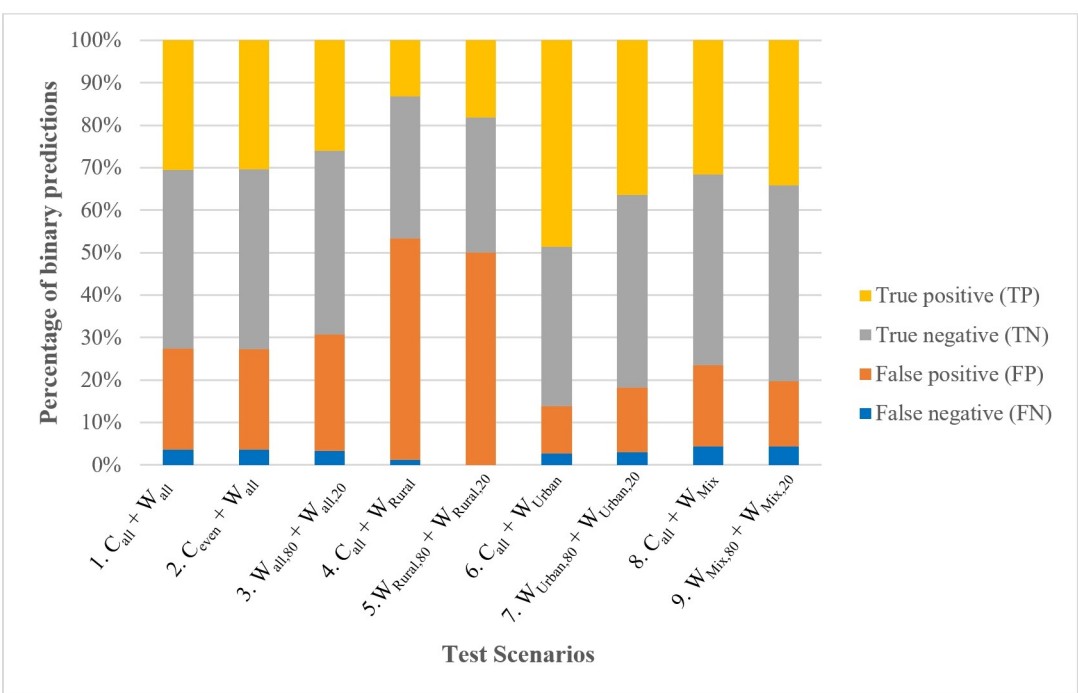

**Fig 4. Results of predictions from SVM model under various scenarios; the 'positive' and 'negative' indicate correct prediction by the model, while 'false positive' refers to observed negative predicted as positive, and 'false negative' indicates observed positive predicted as negative.**

It was noted that across scenarios, the percentage of false positive predictions (26.4±14.9%) was significantly higher than those of false negative predictions (2.9±1.5%) ($p<0.05$, independent-sample T-test) (**Fig 4**). This was in agreement with spectra results that indicated that a considerable number of MPN-PCR negative samples (42.4%) also contained two distinctive local peaks at round 540-542nm and 575-576nm (**Fig 2**); and were assigned a positive prediction by the ML models. However, the overall level of false negatives suggested that the predicted results were conservative.

By applying the models based on spatial differences (*e.g.*, Scenarios 4–9, **Table 3**) it was observed that the poorest model performance was found for samples derived from the Rural catchment. The results from this location indicated that the average ACC was <0.6 for all the models when the controls were used to train the model (*i.e.*, **Scenario 4 $C_{all}$ + $W_{Rural}$**, **Table 3**) and when training and testing were conducted using the water samples (**Scenario 5 $W_{Rural,80}$ + $W_{Rural,20}$**, **Table 3**). In fact, the samples for this site were characterized as having a low probability of containing *Campylobacter* (38 positive vs 226 negative observations). Thus, the dataset was substantially biased towards negative results, which may lead to the high instability of the models. In Scenario 5, where the imbalanced samples of positive and negative observations were used for both training and testing, the models had up to 78.3% FNR. Thus, it is suggested that the pure water samples not to be used for model training and testing.

The best performance was observed for the urban catchment site, which had an almost equal number of positive and negative observations (96 vs 91). It should be noted that, in contrast to the rural catchment, the highly urbanized catchment has inputs from local stormwater infrastructure and is significantly impacted by surface run-off events. Therefore, microbiota captured within enrichment cultures were expected to significantly differ from those observed in more rural/agricultural locations. The average ACC was 0.830, and FNR was 9.8% across all

three models and two scenarios (**Scenario 6 and 7**). The models also demonstrated satisfactory performance when fed with data from the mixed catchment, with an average ACC of 0.769 and FNR of 13.5% (**Scenario 8 $C_{all}$ + $W_{Mix}$**), which was better than **Scenario 3 $W_{all,80}$ + $W_{all,20}$** which used the whole data set (ACC = 0.731 and FNR = 16.8%; Table 3). It should also be noted that the mixed catchment had the largest number of data points (N = 1089). Thus, it is likely that the model performance on the whole dataset was largely influenced by data derived from this location. By using all the mixed catchment data for model training and testing (*i.e.*, without controls, **Scenario 9 $W_{Mix,80}$ + $W_{Mix,20}$**), overall ACC can be slightly improved to 0.807 (when compared to **Scenario 8 $C_{all}$ + $W_{Mix}$**). However, the FNR also increased to 14.9%. This further suggested that it was sufficient to apply laboratory-prepared controls to train the model, which can then be used to predict environmental water samples.

### 3.3. Performance of the MPN-Spectro-ML method in predicting MPN

Using the predicted presence of *Campylobacter* in each enriched culture, the quantification of the concentration (MPN/L) was found to be variable and dependent on the model applied (Fig 5). The NSE values were, in general, below 0.20 (*i.e.*, poor model performance) and, in many cases, were negative. Overall, the worst performance was observed for Scenario **3**, probably due to its slightly higher FNR (Table 3). The highest accuracy and lowest FNR were simulated for the Urban catchment (**Scenario 6 $C_{all}$ + $W_{Urban}$**, Table 3). Thus, it also had the best predicted MPN values when compared to the observed MPN values (using the MPN-PCR method), with an NSE of up to 0.44. However, equivalent results were not obtained for the Mixed catchment (**Scenario 8 $C_{all}$ + $W_{Mix}$**). Overall, using probability estimates to compute MPN values resulted in better model performance with positive NSE values across all the scenarios (with the exception of **Scenario 3**), and SVM and logistic regression often have relatively better performance than RF.

From the perspective of NSE values, the estimations of MPN using predicted data were poor. Nevertheless, it was noted that there are also large uncertainties within the computation

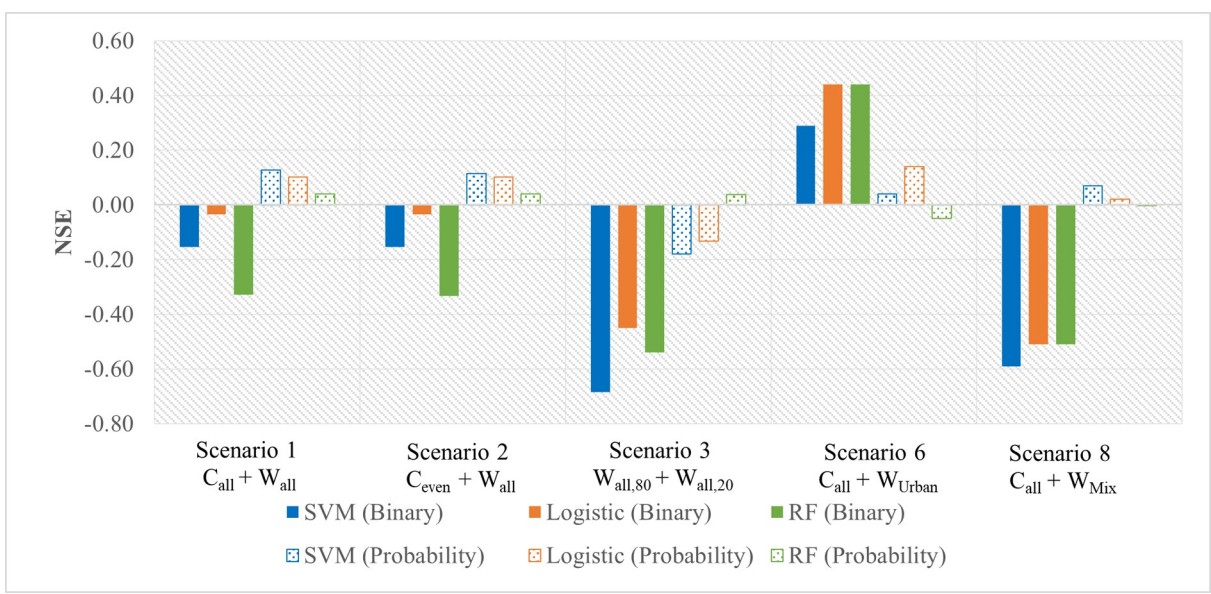

**Fig 5. Performance of the modelling results based on the MPN-Spectro-ML method for estimating MPN values as compared to the measured MPN based on the MPN-PCR method.**

of MPN values. MPN estimates have been reported to be inaccurate for a small number of tubes [37] and highly variable [38]. This can be observed in the comparisons between MPN values estimated from the MPN-Spectro-ML and MPN-PCR method (**Fig 6**). As shown, the confidence intervals of the observed MPN values (green error bars) generally match with the confidence intervals of the predicated MPN values (orange error bars) based on ML methods. Further, inherent uncertainties have been associated with the application of MPN-based quantification methods. Many of these have been previously reviewed [39, 40] but can include the use of non-exact MPN calculations, Type A and Type B uncertainty estimates. Consequently, results are reported as a mean concentration with large associated confidence intervals.

### 3.4. Practical implications and future work

This study highlights the potential of utilizing spectrophotometry for interim reporting of the presence and levels of *Campylobacter spp.* in water systems. When complementing traditional and currently approved methods, this approach can provide regulators with a means to implement interim risk mitigation strategies, resulting in reduced turnaround times and associated costs. Further, given the costs associated with molecular-based technologies, the use of cheaper spectrophotometric methods increases the potential applications of the described technique to resource-poor settings, where there is a large burden of disease associated with environmental transmission of pathogens such as *Campylobacter*.

This study shows significant correlations ($r > 0.40$, $p<0.01$) between *Campylobacter* presence/levels and the absorbance of 56 wavelengths in the range of 531 nm to 586 nm), despite the presence of other microorganisms in the environmental samples collected in this study. Nevertheless, it is possible that other microorganisms present in the environmental samples may produce similar spectral bands and be confused with *Campylobacter*. Thus, future studies could investigate the potential for microbiota-specific effects.

The results suggest that the laboratory-prepared positive and negative controls could provide basic data for training the ML models, which showed relatively acceptable performance in predicting *Campylobacter* presence in various environmental water samples from catchments of different land uses. It is recommended that this new approach be tested considering a wider range of environmental samples and catchments across different regions and climates. While

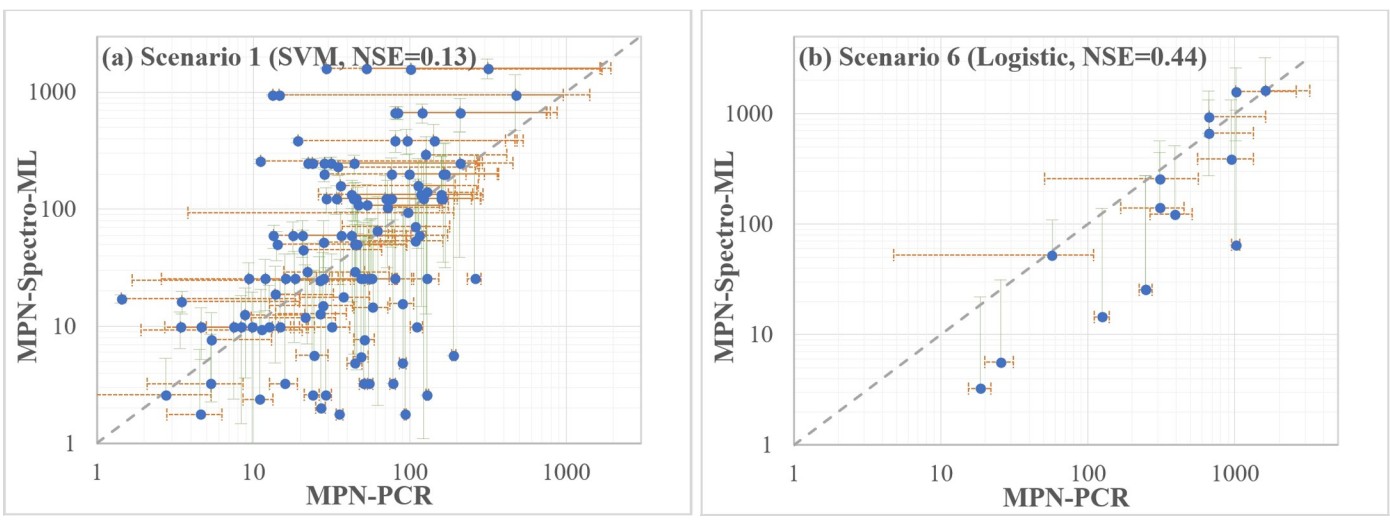

**Fig 6. Comparison of estimated MPN values between MPN_PCR and MPN-Spectro-ML methods.** Error bars indicate the low and high confidence interval of the two methods (orange for MPN-PCR, and green for MPN-Spectro-ML method). Scenario 1 refers to $C_{all} + W_{All}$ and Scenario 6 refers to $C_{all} + W_{Urban}$.

this study tested three ML models, it could also expand to include more ML approaches that can handle various types of data from different environmental conditions.

Given the growing evidence of environmental campylobacter's impact on public health, especially in low- and middle-income countries [41]. Thus, the affordability of spectrophotometry emerges as a key strength, and the output of this study really offers the first steps to a cheap public health response tool with broad applications.

## 4. Conclusions

This study proposed a rapid *Campylobacter* detection method (MPN-Spectro-ML) based on spectrophotometry and machine learning, for application to diverse water matrices. Three machine learning models, namely support vector machine (SVM), logistic regression (LR) and random forest (RF) were used to link the spectrum data with the presence of *Campylobacter*, which was consequently used to estimate the most probable numbers (MPN). This method was then applied to estimate the concentration of *Campylobacter* within the test samples and compared against the traditional MPN-PCR methods. Key results included:

- By analyzing the full spectrum absorbance data, two distinctive local peaks (at 540-542nm and 575-576nm) were observed within >92% of culturally confirmed positive samples.

- Across all different model testing scenarios, similar performance was observed between the three ML models, with an overall prediction accuracy (ACC) of 0.728 and a false negative rate of 6.3%.

- Laboratory controls are recommended for training the models instead of using collected water samples for both training and testing. The trained models could then be used to predict real water samples.

- The MPN of *Campylobacter* estimated based on the new MPN-Spectro-ML method was aligned but not perfectly correlated with that calculated according to the MPN-PCR method (max NSE = 0.44 for the dataset of urban catchment site). Nevertheless, the MPN values based on these two methods were still comparable, considering the confidence intervals and large uncertainties associated with MPN estimation.

## Supporting information

**S1 File. Supporting information.**
(DOCX)

**S2 File. Correlation results.**
(XLSX)

## Acknowledgments

Dr Rhys Coleman, Dr Nick Crosbie, and Dr Melita Stevens are also greatly acknowledged for providing constructive feedback to the manuscript.

## Author Contributions

**Conceptualization:** Kefeng Zhang, David McCarthy.

**Data curation:** Kefeng Zhang, Christelle Schang, Rebekah Henry.

**Formal analysis:** Kefeng Zhang, Christelle Schang, Rebekah Henry.

**Funding acquisition:** Kefeng Zhang, David McCarthy.

**Investigation:** Kefeng Zhang, Christelle Schang, Rebekah Henry.

**Methodology:** Kefeng Zhang, Christelle Schang, Rebekah Henry.

**Project administration:** Kefeng Zhang, David McCarthy.

**Resources:** Kefeng Zhang, David McCarthy.

**Software:** Kefeng Zhang.

**Supervision:** Rebekah Henry, David McCarthy.

**Validation:** Kefeng Zhang.

**Visualization:** Kefeng Zhang.

**Writing – original draft:** Kefeng Zhang.

**Writing – review & editing:** Christelle Schang, Rebekah Henry, David McCarthy.

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
