## [Decision Letter · Decision Letter 0]

26 Apr 2024

PONE-D-24-07765A machine learning approach for rapid early detection of Campylobacter spp. using absorbance spectra collected from enrichment culturesPLOS ONE

Dear Dr. Zhang,

Thank you for submitting your manuscript to PLOS ONE. After careful consideration, we feel that it has merit but does not fully meet PLOS ONE’s publication criteria as it currently stands. Therefore, we invite you to submit a revised version of the manuscript that addresses the points raised during the review process.

Please see below the comments and suggested MAJOR revisions made by the individual(s) who reviewed your manuscript.  If provided, the referee's report(s) indicate the revisions that need to be made before it can be accepted for publication.

We look forward to receiving your revised manuscript.

Kind regards,

Ricardo Santos

Academic Editor

PLOS ONE

“The first and corresponding author (Kefeng Zhang) is supported by Australian Research Council Discovery Early Career Researcher Award (ARC DECRA, DE210101155). The data collected and used in this paper were collected as part of two different Australian Research Council Linkage Projects (LP120100718, LP160100408) and another ARC DECRA (DE140100524). The authors would like to acknowledge Melbourne Water and EPA Victoria for co-funding of the ARC LPs.”

“The first and corresponding author (Kefeng Zhang) is supported by Australian Research Council Discovery Early Career Researcher Award (ARC DECRA, DE210101155). The data collected and used in this paper were collected as part of two different Australian Research Council Linkage Projects (LP120100718, LP160100408) and another ARC DECRA (DE140100524). The authors would like to acknowledge Melbourne Water and EPA Victoria for co-funding of the ARC LPs. Dr Rhys Coleman, Dr Nick Crosbie, and Dr Melita Stevens are also greatly acknowledged for providing constructive feedback to the manuscript.”

“The first and corresponding author (Kefeng Zhang) is supported by Australian Research Council Discovery Early Career Researcher Award (ARC DECRA, DE210101155). The data collected and used in this paper were collected as part of two different Australian Research Council Linkage Projects (LP120100718, LP160100408) and another ARC DECRA (DE140100524). The authors would like to acknowledge Melbourne Water and EPA Victoria for co-funding of the ARC LPs.”

5. PLOS requires an ORCID iD for the corresponding author in Editorial Manager on papers submitted after December 6th, 2016. Please ensure that you have an ORCID iD and that it is validated in Editorial Manager. To do this, go to ‘Update my Information’ (in the upper left-hand corner of the main menu), and click on the Fetch/Validate link next to the ORCID field. This will take you to the ORCID site and allow you to create a new iD or authenticate a pre-existing iD in Editorial Manager. Please see the following video for instructions on linking an ORCID iD to your Editorial Manager account: https://www.youtube.com/watch?v=_xcclfuvtxQ.

Reviewers' comments:

Reviewer's Responses to Questions

**Comments to the Author**

1. Is the manuscript technically sound, and do the data support the conclusions?

Reviewer #1: Partly

2. Has the statistical analysis been performed appropriately and rigorously? 

Reviewer #1: No

3. Have the authors made all data underlying the findings in their manuscript fully available?

Reviewer #1: Yes

4. Is the manuscript presented in an intelligible fashion and written in standard English?

Reviewer #1: Yes

5. Review Comments to the Author

Reviewer #1: Dear authors,

thank you for submitting your research, it is very interesting and relevant work you do!

Overall, I like your submission. You explore the potential of ML methods improving the speed and potentially the quality of the assessment of probes for the presence of Campylobacter.

The introduction to the topic is clear and concise, so is the description of the employed data collection and ML methods.

As my expertise lies in the ML area, I will focus my comments on the corresponding parts of the submission. To summarize, I have major concerns with the training, testing and evaluation procedures of the ML methods, and a list of minor comments for improving the manuscript.

Major: ML methods

It is good to start with simple methods like logistic regression, SVM and RF. However, the evaluation should be thorough. Most importantly, it was not entirely clear how the training and hyperparameter selection was performed. To select the hyperparameters, one should perform a N-fold crossvalidation for each hyperparameter, e.g., split the training data into 80-20 random splits and train and test on different splits for each hyperparameter, then select the one with the lowest average test performance.

After this selection process, the method should be trained on the entire training data set again using the selected hyperparameters and then finally applied to the actual test set which was not used at all before.

It is essential the report error bars on the report classifier performances (Table 3). To that end, one could repeat the above procedure 5-10 (N) times and report mean test performance +- standard error of the mean (mean +- std / sqrt(N).

What is the dimensionality of the spectral data? If it is high, this might be an explanation for the relatively low performance of the classifiers. It might be worth exploring non-linear methods, e.g., neural-network based logistic regression. However, I see that the training data is limited and might not suffices for training neural networks.

Further questions and questions:

1) How exactly is the MPN calculated? In line 170 you mention “by applying probability theory”. I think this part definitively need more explanation, given that it seems to be the central assessment for the presence of Campylobacter. I am also wondering what it is needed at all. Why not report the presence of the bacteria as binary variable and take the average over probes?

2) How exactly is the correlation between spectral data (continuous, high-dimensional) and Campylobacter presence (binary) calculated?

3) RF should also use the class_weight="balanced" option because the number of positive and negative examples is not equal.

4) Table 3: the extraordinary high training accuracy is a strong sign for overfitting! Also, how can the test accuracy be 1.0 for test 5? This looks suspicious. Error bars would help here (see comment above).

5) Figure 5: Absolute numbers are difficult to interpret. Numbers should be relative, e.g., percentage or proportions.

6) The justification for using NSE for comparing the MPN predictions is not clear. Also, the explanations for how exactly NSE and the confidence intervals are calculated are missing. As a consequence, Figures 5 and 6 are unclear, e.g., why can NSE be negative and  what does that mean?

Minor comments:

line 15: first sentence seems grammatically off and difficult to parse

line 39: “-a”

line 49: change of font size.

line 58: MPN-PCR has not been introduced

line 60: AS/NZS not introduced

line 85: acronyms not introduced

line 89: Argument is not sound. How does this sentence follow from the previous sentence?

line 107: Typo “these”, “spp”

line 199: Unlikely that this is the correct reference for logisitic regression or the sigmoid function.

line 420: what are “good” correlations? I suggest to use a different more objective adjective.

Overall, I think that the ML training and evaluation part needs significant changes and more explanations. But once this is addressed, I think that the paper would be a valuable contribution.

6. PLOS authors have the option to publish the peer review history of their article (what does this mean?). If published, this will include your full peer review and any attached files.

Reviewer #1: No

---

## [Author Response · Author response to Decision Letter 0]

10 Jun 2024

Please refer to the “Response to Reviewers” document for our point-to-point responses to each comment.

---

## [Decision Letter · Decision Letter 1]

9 Jul 2024

A machine learning approach for rapid early detection of Campylobacter spp. using absorbance spectra collected from enrichment cultures

PONE-D-24-07765R1

Dear Dr. Zhang,

We’re pleased to inform you that your manuscript has been judged scientifically suitable for publication and will be formally accepted for publication once it meets all outstanding technical requirements.

Kind regards,

Ricardo Santos

Academic Editor

PLOS ONE

Additional Editor Comments (optional):

Reviewers' comments:

Reviewer's Responses to Questions

**Comments to the Author**

1. If the authors have adequately addressed your comments raised in a previous round of review and you feel that this manuscript is now acceptable for publication, you may indicate that here to bypass the “Comments to the Author” section, enter your conflict of interest statement in the “Confidential to Editor” section, and submit your "Accept" recommendation.

Reviewer #1: All comments have been addressed

2. Is the manuscript technically sound, and do the data support the conclusions?

Reviewer #1: Yes

3. Has the statistical analysis been performed appropriately and rigorously? 

Reviewer #1: Yes

4. Have the authors made all data underlying the findings in their manuscript fully available?

Reviewer #1: Yes

5. Is the manuscript presented in an intelligible fashion and written in standard English?

Reviewer #1: Yes

6. Review Comments to the Author

Reviewer #1: Dear authors, thank you for submitting a revised version of the manuscript! The point-to-point response to my comments is clear and addresses all the questions and concerns I had with the initial submission.

I have two more comments:

1) I want to encourage you to also make the code available online, e.g, on GitHub, to make your study more accessible and easier to reproduce.

2) I agree with you that your dataset is very interesting and relevant. I think it would be great to provide the dataset and the task description for predicting the presence of the bacteria on a competition platform like Kaggle. This will give you access to a large community of user that will try to solve the task in different ways, giving you inspiration for your future work on this topic.

Best wishes,

Jan

7. PLOS authors have the option to publish the peer review history of their article (what does this mean?). If published, this will include your full peer review and any attached files.

Reviewer #1: No

---

## [Editor Report · Acceptance letter]

19 Jul 2024

PONE-D-24-07765R1 

PLOS ONE

Dear Dr. Zhang, 

I'm pleased to inform you that your manuscript has been deemed suitable for publication in PLOS ONE. Congratulations! Your manuscript is now being handed over to our production team.

Kind regards, 

on behalf of

Dr. Ricardo Santos 

Academic Editor

PLOS ONE